# Polycystic Ovary Syndrome and the Internet of Things: A Scoping Review

**DOI:** 10.3390/healthcare12161671

**Published:** 2024-08-21

**Authors:** Sandro Graca, Folashade Alloh, Lukasz Lagojda, Alexander Dallaway, Ioannis Kyrou, Harpal S. Randeva, Chris Kite

**Affiliations:** 1School of Health and Society, Faculty of Education, Health and Wellbeing, University of Wolverhampton, Wolverhampton WV1 1LY, UK; s.graca@wlv.ac.uk (S.G.); f.alloh@wlv.ac.uk (F.A.); alex.dallaway@wlv.ac.uk (A.D.); 2Department of Nursing Sciences, Faculty of Health & Social Sciences, Bournemouth University, Fern Barrow, Poole BH12 5BB, UK; 3Warwickshire Institute for the Study of Diabetes, Endocrinology and Metabolism (WISDEM), University Hospitals Coventry and Warwickshire NHS Trust, Coventry CV2 2DX, UK; l.lagojda@sheffield.ac.uk (L.L.);; 4Clinical Evidence Based Information Service (CEBIS), University Hospitals Coventry and Warwickshire NHS Trust, Coventry CV2 2DX, UK; 5Sheffield Centre for Health and Related Research, School of Medicine and Population Health, University of Sheffield, Sheffield S1 4DA, UK; 6Warwick Medical School, University of Warwick, Coventry CV4 7AL, UK; 7Centre for Sport, Exercise and Life Sciences, Research Institute for Health & Wellbeing, Coventry University, Coventry CV1 5FB, UK; 8Institute for Cardiometabolic Medicine, University Hospitals Coventry and Warwickshire NHS Trust, Coventry CV2 2DX, UK; 9Aston Medical School, College of Health and Life Sciences, Aston University, Birmingham B4 7ET, UK; 10College of Health, Psychology and Social Care, University of Derby, Derby DE22 1GB, UK; 11Laboratory of Dietetics and Quality of Life, Department of Food Science and Human Nutrition, School of Food and Nutritional Sciences, Agricultural University of Athens, 11855 Athens, Greece; 12Chester Medical School, University of Chester, Shrewsbury SY3 8HQ, UK

**Keywords:** polycystic ovary syndrome (PCOS), Internet of Things (IoT), mobile app, social media, wearable, machine learning, artificial intelligence (AI)

## Abstract

Polycystic ovary syndrome (PCOS) is a prevalent endocrine disorder impacting women’s health and quality of life. This scoping review explores the use of the Internet of Things (IoT) in PCOS management. Results were grouped into six domains of the IoT: mobile apps, social media, wearables, machine learning, websites, and phone-based. A further domain was created to capture participants’ perspectives on using the IoT in PCOS management. Mobile apps appear to be useful for menstrual cycle tracking, symptom recording, and education. Despite concerns regarding the quality and reliability of social media content, these platforms may play an important role in disseminating PCOS-related information. Wearables facilitate detailed symptom monitoring and improve communication with healthcare providers. Machine learning algorithms show promising results in PCOS diagnosis accuracy, risk prediction, and app development. Although abundant, PCOS-related content on websites may lack quality and cultural considerations. While patients express concerns about online misinformation, they consider online forums valuable for peer connection. Using text messages and phone calls to provide feedback and support to PCOS patients may help them improve lifestyle behaviors and self-management skills. Advancing evidence-based, culturally sensitive, and accessible IoT solutions can enhance their potential to transform PCOS care, address misinformation, and empower women to better manage their symptoms.

## 1. Introduction

Polycystic ovary syndrome (PCOS) is a highly prevalent endocrine disorder with heterogeneous symptomatology (e.g., metabolic, reproductive, and psychological symptoms), which negatively impacts the health-related quality of life (HRQoL) of affected women [1,2,3]. The global prevalence of PCOS has increased in the past decades [4], with an estimated prevalence of 11–13% worldwide [5], and a significant related economic burden on healthcare systems. For example, the estimated economic burden of PCOS in the UK was GBP 950 million in 2023, representing an increase from approximately GBP 237 million in 2014 [6].

The 2023 international evidence-based PCOS guidelines [3] recommend that a PCOS diagnosis should be based on criteria adapted from the Rotterdam criteria [7]; thus, requiring at least two of (1) clinical/biochemical hyperandrogenism; (2) ovulatory dysfunction; and (3) polycystic ovaries on ultrasound (anti-Mullerian hormone can be used instead of ultrasound) [3]. Based on these diagnostic criteria, there are different PCOS phenotypic groups that are further influenced by both non-modifiable (e.g., genotype, ethnicity, and life stage) and modifiable (e.g., lifestyle and body weight) [8] factors. Whilst not part of the diagnostic criteria, insulin resistance is acknowledged as an additional key factor of the PCOS pathophysiology, which typically exacerbates many aspects of the syndrome [3,8].

Given the lack of definite PCOS treatment, individuals with PCOS have expressed the preference for multi-disciplinary, integrated care by health professionals with expertise in PCOS who will listen and help them to focus on practical skills when providing lifestyle advice, ultimately empowering them to self-manage their condition [9,10]. Reported gaps in knowledge among health professionals [11,12,13,14] are contributing to diagnostic delays and variations in the care provided, perpetuating patient dissatisfaction and unmet information needs [15,16,17,18]. Combined with a paucity of easily accessible and evidence-based information, this is leading patients to seek help through the internet, often from sources that are of low quality, unaccredited, inaccurate, and/or commercially driven [16].

Along with the internet, social media, mobile apps, and wearables have also been identified as having an emerging clinical role in healthcare [19,20]. For example, wearables are increasingly used in chronic disease management [21], smart maternal healthcare [22], and to increase physical activity [23]. Due to technological advancements and increased awareness, technology-driven solutions specifically designed to address women’s health needs continue to proliferate, with predictions estimating that the “femtech” market will be worth USD 50 million by 2025 [24].

The Internet of Things (IoT) comprises the ever-evolving multitude of internet-enabled technologies and devices that are capable of sensing, collecting, and processing information around them in order to then communicate it by networking with each other or with other devices and services using the internet [25]. Within such an ecosystem of smart, interconnected technology, there are instances when “the connected thing” extends beyond physical objects to a biological entity, such as a person, using sensors like a glucose monitor, activity tracker, or heart monitor implant [26]. Through the use of body sensors and extensive processing capabilities [20,27], wearables can range from personal use (e.g., smartwatches or fitness trackers) [28] to devices that monitor physiological parameters (e.g., heart rate, blood glucose, and blood pressure), as well as multi-functional health examination instruments (e.g., connecting the obstetrician to a pregnant woman using IoT technology) [22]. Thus, despite inherent challenges related to patient privacy and data security, there is immense potential for researchers and clinicians to use the IoT in healthcare [26,29].

Based on the aforementioned evolving landscape of the IoT in healthcare, the aim of the present scoping review is to explore the use of the IoT in the context of PCOS. As such, the primary objectives of this review are to identify the users, processes, and platforms used, as well as the main findings, the behaviors, and the reported outcomes and experiences of those using the IoT for PCOS-related reasons. This can help to identify current gaps in optimizing its use, support further development and technological integration, inform future research, and guide those with PCOS toward self-management solutions to improve patient outcomes.

Note on Gender Sensitive Language and Inclusiveness

The authors echo the statement for inclusiveness and stigma in the international evidence-based guideline for the assessment and management of PCOS [3,30]. Striving to avoid miscommunication or contributing to the erasure of gender terms “woman/women” and acknowledging and respecting gender identities, diversity, and inclusivity, the authors will follow the language used in the 2023 PCOS guideline [3,30,31].

## 2. Materials and Methods

This scoping review was conducted following the methodology proposed by Arksey and O’Malley [32] and further developed by Levac et al. [33]. As detailed in the following sections, after defining the research question, we identified relevant studies and proceeded to study selection based on predefined eligibility criteria. Data were then extracted and charted in order to be summarized. Rather than focusing on the sixth step of the methodology regarding consulting key stakeholders [32], we instead followed the updated framework by the Joanna Briggs Institute (JBI) Scoping Review Methodology Group and looked at possible gaps, implications of the findings, and suggestions for future work [34,35]. The reporting of this scoping review was also guided by the Preferred Reporting Items for Systematic Reviews and Meta-Analyses extension for Scoping Reviews (PRISMA-ScR) checklist [36] (Appendix A).

### 2.1. Research Question

The research question guiding this scoping review was developed to incorporate the elements of the PCC (Population, Concept, Context) framework [35]: what and how is the IoT (concept) being used in the management (context) of women with PCOS (population)? To help outline and map the findings focusing on practical applications and ecological validity, this scoping review also set out to explore and present the characteristics, reported reasons, behaviors, and experiences of the users, alongside the processes, platforms, and reported outcomes.

### 2.2. Eligibility Criteria

The eligibility criteria were informed by team discussions, previous work, and preliminary searches and were consequently adapted from the PCC framework [35]. No time range limit and no language restriction were applied for articles published in peer-reviewed journals available as full-text. The inclusion criteria extend to publications describing the development, setup, or use of the IoT for or by people of any age diagnosed with PCOS. This included IoT-related devices capable of collecting health metrics (e.g., wearables such as activity trackers and sleep monitors), as well as those capable of storing, sharing, or accessing personalized data (e.g., mobile phones and mobile apps, such as menstrual cycle and symptom trackers), along with social media (e.g., WeChat, YouTube, and Facebook groups), and overall internet content (e.g., search engines, patient support groups, and internet forums). Artificial intelligence (AI) and machine learning processes used in the development or running of any PCOS-related systems and platforms (e.g., chatbots and data analysis for diagnosis prediction) were also included. Studies that did not include people with PCOS and studies that did not either examine the use of the IoT in PCOS or evaluate the effectiveness of the IoT as part of an intervention (e.g., wearables) were excluded.

### 2.3. Search Strategy—Information Sources

The search strategy was drafted through team discussion and further refined by LL (Appendix A). Searches in Medline and IEEE were completed in December 2023. Search results were exported to the review manager software Rayyan [37], where duplicates were removed, and independent title and abstract screening was completed by SG, FA, and CK. Disagreements were resolved by discussion (SG and CK) and full texts were screened by SG.

### 2.4. Data Extraction and Analysis

Data charting followed a standardized data extraction form [38] and allowed for refinements according to information emerging during the review stage, as the iterative process of data extraction in scoping reviews is conducive to this [39]. Data analysis was primarily descriptive in nature, as scoping reviews are exploratory and aim to identify the extent of current available evidence and summarize it, rather than synthesize results [32,39]. Along with frequency counting, a descriptive qualitative approach was favored for retrieved qualitative studies, using basic coding of data to organize these into categories, focusing on identifying characteristics, definitions, and concepts pertinent to the review question [38,39].

## 3. Results

Database searches returned a total of 219 studies of which only two were identified as duplicates. Title/abstract screening of the remaining 217 publications resulted in the exclusion of 145 papers. Following this, the full text of 72 articles were retrieved for appraisal, resulting in the exclusion (with reasons) of a further eight papers (Figure 1). This left 64 publications which met the eligibility criteria. When publications were linked to the same study, they were grouped together for reporting, resulting in 59 eligible studies. The 12 linked publications were: Alotaibi and Shaman 2020 [40] with Alotaibi and Alsinan 2016 [41]; Lee and Lee 2023 [42] with Choi et al., 2023 [43]; Lim et al., 2021 [44] with Ee et al., 2020 [45]; Chiu et al., 2018 [46] with Htet et al., 2018 [47]; and Dietz de Loos et al., 2022 [48] with Jiskoot et al., 2020 [49], Jiskoot et al., 2020 [50], and Jiskoot et al., 2017 [51].

Given the scope of the IoT, eligible studies were grouped into six domains, consisting of mobile apps (n = 18) [42,43,52,53,54,55,56,57,58,59,60,61,62,63,64,65,66,67], social media (n = 12) [40,41,60,68,69,70,71,72,73,74,75,76], wearables (n = 11) [62,77,78,79,80,81,82,83,84,85,86], machine learning (n = 10) [62,63,64,65,66,67,76,87,88,89], websites (n = 8) [46,47,90,91,92,93,94,95], and mobile phones (n = 6) [48,49,50,51,61,96], with nine studies mentioning multiple domains of the IoT [60,61,62,63,64,65,66,67,76]. A seventh domain based upon qualitative studies reporting on participants’ voices (n = 9) [17,44,45,97,98,99,100,101,102] was created to capture participant perspectives on using the IoT for PCOS management (Figure 2).

The publishing dates ranged from 2007 [102] to 2023 [73], with the majority (n = 42) being published since 2020, especially in 2023 (n = 23). The highest number of included studies originated from the USA (n = 11, 17.2%) and Australia (n = 10, 15.6%). Eight studies (12.5%) were conducted in the UK, seven (10.9%) each from China and India, and four (6.3%) from The Netherlands. There were also two studies (3.1%) each from Brazil, Canada, Croatia, New Zealand, Korea, and Saudi Arabia, and one study (1.6%) each from France, Greece, Jamaica, Poland, and Turkey.

### 3.1. Mobile Apps

In total, 13 studies included information regarding the use of mobile apps [42,43,52,53,54,55,56,57,58,59,60,61,62]. Five other studies that mention mobile apps were described in the overall context of machine learning [63,64,65,66,67] (Appendix A). Study sample sizes ranged from 28 [42] up to 416,712 [56], although it should be noted that the latter are drawn from an international sample where the majority of participants do not have PCOS. Where reported, the mean age (±standard deviation) of participants ranged from 24.70 ± 5.45 years [56] to 33.0 ± 8.2 years [61], whilst the mean body mass index (BMI) ranged from 24.2 kg/m^2^ [43] to 29.3 ± 8.00 kg/m^2^ [61].

Two studies from Australia relate to the translation of the PCOS guidelines and the development of its accompanying app, AskPCOS [58,59]. One of them reported that of the respondents to a national survey (n = 264), the vast majority (86%) would be more likely to use an app than a website (14%) [58]. Furthermore, 91% stated that they would use a PCOS-specific app, should it be available to them [58]. Evidence-based information (95%), ability to record symptoms (95%), and opportunities to ask questions to an expert (86%) were features the respondents considered important for such a PCOS-specific app [58].

Other apps used in the included studies were Flo [55,56], Read Your Body [52], Mint Health [60], PCOS Monitoring System [62], and Home of PCOS [57]. One study did not disclose the name of the Android app they developed [43] and consequently used it in their trial [42]. Menstrual tracking apps (i.e., Flo and Read Your Body) were the most commonly used apps in studies including participants with and without PCOS conducted in New Zealand (n = 36, 51%) [54] and USA (n = 78, 21.19%) [53], respectively. In the former, participants (n = 144, aged 19 to 55 years) reported having previously used a menstrual app (n = 71, 49%) to assist in the management of menstrual disorders and to keep a record of menstrual cycle dates and their symptoms [54]. In the latter, participants (n = 368, aged 20 to 49 years) reported that their most frequently used technology was a urine hormone test or monitor (n = 299, 81.3%), mobile app (n = 253, 68.8%), or temperature tracking device (n = 116, 31.5%) [53]. Of those with PCOS (n = 55, 14.9%), the majority (n = 35, 63.6%) reported that the use of tracking technologies helped lead to their reproductive disorder diagnoses [53]. A similar opinion was shared by those with endometriosis (n = 22, 62.8%) and infertility (n = 15, 75%) [53].

One study reported data from 416,712 users who initiated the PCOS chatbot dialogue on the mobile app Flo from the USA (n = 243,238), UK (n = 68,325), India (n = 40,092), Philippines (n = 35,131), and Australia (n = 29,926), aged between 24.7 ± 5.45 and 29.7 ± 5.97 years [56]. The most prevalent predictors of PCOS were bloating, both high cholesterol and glucose, and high glucose alone [56].

Studies from China [57] and Korea [42] showed how the use of mobile apps contributed to decreased body weight/BMI, waist circumference, anxiety, and depression scores. For example, a 4.4% weight loss (75.84 to 72.65 kg) after 12 weeks was noted compared to 1.1% (72.98 to 72.19 kg) in the control group [42], with the intervention group also showing significant improvements in postprandial insulin levels, hirsutism, and depression [42].

### 3.2. Social Media

Ten studies included information regarding the use of social media [40,41,68,69,70,71,72,73,74,75]. Two other studies that mention social media were described in the overall context of mobile apps [60] and machine learning [76] (Appendix A). Most of the included studies (n = 6) were published in 2023 [68,69,71,72,73,74], and two were published in 2022 [70,75]. A 2022 study conducted in China using the Mint Health app also used WeChat to educate patients on PCOS and support them with self-management methods, encouraging them to be more actively involved in treatment decisions and interventions [60]. A 2023 study was conducted in New Zealand using machine learning to explore the feasibility of gathering and analyzing a dataset of self-reported laboratory test results posted on the PCOS subreddit on Reddit [76]. Two studies originating from Saudi Arabia published in 2016 and 2020 were grouped together for analysis [40,41]. These are related to the system architecture of a region-specific private social media platform called “Mobile PCOS Management and Awareness System for Gulf Countries” [41] and the subsequent trial using it [40].

The most recent of the included studies originated from the USA, and it was designed to assess the content, engagement, and extent of PCOS-related information on TikTok, Instagram, and Reddit [73]. It reported an average of 1.8 million views for PCOS-related content on TikTok alone [73]. In samples of 100 posts each, a conflict of interest, such as advertising the sale of supplements or health coaching sessions by the influencer, was present in 45% of TikTok and 89% of Instagram posts [73]. On TikTok, weight and ovarian cysts were mentioned as the most and the least mentioned topics, respectively, while on Instagram, diet was the most commonly mentioned topic, and oral contraceptive pills the least. On Reddit (n = 22,641 posts), the most comments were found on posts mentioning “symptom management” and “community experiences”, whereas posts related to “weight management”, “healthcare providers”, and “general questions” received the fewest comments [73]. Two studies from China used WeChat to provide lifestyle advice [74,75]. For those with PCOS undergoing assisted reproductive technology treatment, the advice through WeChat helped them improve their self-management skills, especially weight-controlling behaviors, and enhance oocyte quality [75]. Followers of the WeChat public account “Fan says women’s health” appeared to maintain a healthier lifestyle in both dietary and physical activity assessments [74].

Two UK studies analyzed data from Twitter/X to establish the demographics and experiences of the top 100 PCOS “influencers” and organizations advocating for PCOS [70] and to study the digital impact of the annual initiative “PCOS Awareness Month” [71]. The majority of the top 100 “influencers” were women (73.2%) and from high-income countries (95%), predominantly the USA (n = 49) and the UK (n = 22) [70]. Similarly, 80% of the top 100 organizations operated in high-income countries, mainly the USA (n = 38) and the UK (n = 27) [70]. Seven of the interviewed “influencers” (n = 8) named the spread of misinformation as their motivation and reason why they decided to become involved with PCOS awareness [70]. Coinciding with the increased global online activity due to the COVID-19 pandemic and consequent lockdowns, September 2020 saw the highest spike of total tweets (n = 16,465), constituting the highest yearly increase (136.1%) in total tweets since September 2014 [71]. The top 10 most influential accounts in 2021 and 2022 were divided between PCOS researchers and/or advocates and organizations, with seven of those users being the same in both years [71]. Most engagement was reported in the USA, UK, Australia, India, Canada, South Africa, and Trinidad & Tobago, while there was limited engagement in African, Asian, South American, and non-English speaking European countries [71].

Three of the included studies, originating from Turkey [68], Jamaica [69], and the UK [72], analyzed YouTube content. The content analysis of 198 YouTube videos containing PCOS-related exercises revealed that uploads from India (n = 91) were almost equal to USA, Canada, and Europe combined (n = 90) [68]. Most of the videos (n = 133, 67.2%) were uploaded after the COVID-19 pandemic, mainly by health employees (n = 28, 14.1%), yet only 8.6% (n = 17) cited a scientific article [68]. The videos were mainly about yoga (n = 58, 29.3%), strength training (n = 44, 22.2%), aerobic exercise (n = 38, 19.2%), or a combination of at least two of those exercises (n = 58, 29.3%) [68]. The main parameters highlighted along with exercise were hormonal balance and/or imbalance (n = 85, 42.9%), dietary recommendations (n = 74, 37.4%), and insulin resistance (n = 71, 35.9%,) [68]. Analysis of 80 PCOS-related YouTube videos with a total of 36,437,534 views found that most were uploaded by non-physicians (n = 30, 37%) and that those uploaded by patients (n = 7, 8.8%) had higher popularity and a lower global quality score than videos uploaded by hospitals (n = 13, 16.3%) [69]. The final study, which looked at the context of comments (n = 85,872) posted on YouTube videos about PCOS (n = 940) across 12 years, found that, where it was possible to identify gender (n = 13,106), 88.5% of those comments (n = 11,601) were posted by female users [72]. The most frequently used keyword was period (n = 9352), and the main associated theme was PCOS symptoms, such as irregular periods and acne [72]. Misinformation regarding the “cure” for PCOS was the key theme associated with comments by male users (n = 1506, 11.5%) [72].

### 3.3. Wearables

Ten studies included information regarding the use of wearables [77,78,79,80,81,82,83,84,85,86] (Appendix A). Publication dates spanned from 2009 to 2022, with three of them originating from China [81,85,86], two from Brazil [78,82], two from the USA [77], and one each from Croatia [84], Poland [83], and the UK [80]. One further study, originating from India and published in 2023, was described in the context of mobile apps [62], since it reported the development of an app using machine learning and Bluetooth connectivity to a galvanic skin response sensor capable of collecting stress level information [62].

Wearables used included activity trackers [78,79,82], sleep monitors [77,80], 24-h blood pressure monitors [83], and more invasive ones inserted into the abdominal interstitial tissue for continuous glucose monitoring [81,84,85,86]. The sample size of included studies ranged from 30 [77] to 151 [78], whilst the mean age of participants ranged from 15.4 years [79] to 35.4 years [77], and the mean BMI ranged from 20.4 kg/m^2^ [86] to 38.1 kg/m^2^ [79].

Studies using wearable sleep sensors found that sleep-disordered breathing was more common among patients with PCOS (87.5% vs. 45.5%) [77], and sleep efficiency was significantly lower for those with PCOS (Actiwatch data: 82.8 ± 4.7 vs. 85.6 ± 4.3) [80].

A study using digital pedometers showed that active women with PCOS had a better anthropometric and metabolic profile than sedentary ones of the same age, whilst an increment of 2000 steps per day in habitual physical activity was independently associated with decreased free androgen index [78]. Armband activity trackers revealed how teens and young adults with PCOS in the USA preferred bouts of activity lasting at least five to ten minutes, but less than 30 min [79]. Adolescents with PCOS, particularly those with obesity, may present asymptomatic adverse alterations (e.g., in blood pressure and resting heart rate), which are considered early cardiovascular disease risk factors [83].

### 3.4. Machine Learning

Nine included studies mention the use of machine learning [63,64,65,66,67,76,87,88,89] (Appendix A). Most of them were published in 2023 (n = 5) [63,64,65,76,87], while two were published in 2022 [88,89], and one each in 2021 [66] and 2020 [67]. The majority of these studies originated from India (n = 5) [63,64,65,66,88], while three originated from the USA (n = 3) [67,87,89], and one from New Zealand [76]. A previously described study in the context of mobile apps mentioned that machine learning was used during the app development process [62].

The utility of AI and machine learning in the diagnosis of PCOS using clinical, genetic, and proteomic data, and electronic health records as data sources was explored in a systematic review [87]. This included 31 studies, which were predominantly from India (29%) or China (16%), with sample sizes ranging from 9 to 2000 and a median age of 29 years [87]. Ultrasound images were used in 48% of them, and only a third (n = 10, 32%) used standardized PCOS diagnostic criteria as a reference to evaluate their results [87]. By comparing data from published literature with laboratory test results posted on the Reddit PCOS subreddit, one study concluded that such data were representative of PCOS research cohorts, suggesting more data collected online could be used for PCOS research [76]. One study developed a prediction model capable of generating a PCOS risk score and tested it by analyzing data from the irregular cycle feature of the app Clue, with the model showing a higher probability of prediction when compared to assessment by a clinician [67]. When machine learning was tested for PCOS diagnosis prediction, the reported accuracy varied between 81% [66] and 100% [88]. The latter used follicular fluid samples, while other studies using patient data reported an accuracy of 90.44% [65], 92.45% [63], and 98.41% [64]. Two of those studies reported that the machine learning model for PCOS diagnosis was consequently included in apps containing menstrual cycle trackers, namely MonAmie, which also provides exercise and fitness advice [64], and BeReddy, which includes a chatbot [65].

### 3.5. Websites

Eight included studies related to PCOS content on websites [46,47,90,91,92,93,94,95] (Appendix A). Their publication dates range from 2010 to 2023. Regarding publication date and country of origin, two studies published in 2020 were from France [90] and USA [92], two in 2018 from Australia [46,47], one in 2016 from the USA [93], one in 2023 from Croatia [91], one in 2012 from Greece [94], and another in 2010 from the UK [95]. The majority (n = 5) involved analysis of content from search engine results [46,47,91,94,95], while one each looked at content in teen and women’s digital magazines [93], French language internet forums [90], and a specific PCOS-related website [92].

Search engine results reported in studies published in 2010 [95] and 2012 [94] included 15 websites each, with all but two Australian websites originating from either the USA or the UK. The 2010 study also revealed that only two of the websites correctly described the criteria for PCOS diagnosis [95]. More recent studies highlighted the high number of websites in their findings with a commercial background [46,47,91], also pointing out a possible advantage in the quality of results when using Google, compared to other search engines [91]. A 2018 study highlighted the paucity of lifestyle information on PCOS websites and how the overall accuracy of the information on websites was suboptimal [47].

Content analysis of French language internet forums revealed how those with PCOS perceive the medical information provided to them as inadequate or insufficient, as well as how anxiety and psychological distress are frequently omitted by their health practitioners [90]. Nevertheless, the results from this study show how discussion forums appear to be beneficial by providing a source of support and sharing of empirical knowledge between those with PCOS [90].

A study looking at 95 randomly selected stories posted by users of a PCOS support website in the USA reported on how significant bio-psychological and socio-cultural aspects of PCOS impact their daily lives and medical experiences, affecting their self-image, coping efficacy, and health outcomes [92]. It also highlighted the need for cultural awareness, providing PCOS education, and innovative solutions to healthcare inequities [92]. Such needs were further accentuated by the results of content analysis of teen and women’s digital magazines, which noted an absence of discourse on race and ethnicity and a paucity of content directed toward Latinas and African American women [93]. This study also noted the underrepresentation of adolescents with PCOS in magazine articles [93].

### 3.6. Phone-Based

Six publications are included in this category (Appendix A). In five of them, either text messages providing feedback to the patients [48,49,50,51], or both text messages and phone calls with the patients are mentioned [96]. One study, previously described in the context of mobile apps, implies the collection of health data and menstrual cycle tracking through apps from iPhone users with an iCloud account [61]. One of the studies is a protocol published in 2023 from India regarding a multicenter trial to evaluate the effectiveness of an individualized lifestyle intervention for women with PCOS who are trying to conceive [96]. This proposes weekly text messages and/or videos on diet and physical exercise, as well as monthly telephone contacts to assess diet and exercise compliance [96]. The other four publications refer to a study conducted in the Netherlands, including the 2022 report on the trial [48], two 2020 papers focusing on the primary outcome [50], and secondary analysis [49], as well as the protocol published in 2017 [51].

### 3.7. Participant Voices

Nine included papers report on patients’ own preferences, views, and experiences of the IoT for PCOS-related reasons [17,44,45,97,98,99,100,101,102] (Appendix A). Table 1 provides an overview of themes identified from the included papers.

The publication dates ranged from 2007 to 2023, with the majority originating from Australia (n = 6) [17,44,45,97,99,102], while two originated from the UK [100,101], and one from Canada [98]. Two of the papers from Australia, published in 2020 [45] and 2021 [44], were reported together as they refer to the same study. Participants reported that living with PCOS generates a level of anxiety, loss of feminine identity, and dissatisfaction with current models of care (n = 10, age: 36.1 ± 7.24 years, BMI: 36.38 ± 7.8 kg/m^2^) [45]. Weight management was reported as a fundamental concern [45], with the authors suggesting an increase in interactive health literacy, focusing on weight management skills guided by peer support within social networks of PCOS patient groups [44].

One of the included studies from the UK, published in 2013, reported how members of a PCOS charity (n = 50, age: 33.6 ± 5.36 years) felt empowered from their participation in an online support group, through connecting and learning from others living with the same condition [101]. The positive effect of connecting with others through social networks was also mentioned in a 2016 UK study (n = 9, age: 20–45 years), and an Australian study published in 2018 (n = 13, age: 22–43, mean age: 30.8 years) reported on positive engagement on a Facebook discussion group [99]. In this study, participants showed a preference for new technologies as a source of information, such as what they considered “trusted websites” on the internet and podcasts with health professionals [99]. The 2007 study originating from Australia (n = 10, age: 28–38, mean age: 32.4 years) reported that those with PCOS expressed their preference to use the internet when seeking information about PCOS mainly due to convenience, privacy, and accessibility [102].

A 2023 study originating from Australia included 1167 participants (age: 32 ± 7 years, BMI: 34.3 ± 8.9 kg/m^2^) predominantly from the USA (70%), reported that 25% and 14% of them sought dietary and physical activity advice from a health professional, respectively [97]. By contrast, more than half relied on the internet and social media as their primary source of dietary (59%) and physical activity (67%) information [97]. Furthermore, a 2022 Australian study capturing the opinions of health care professionals (General Practitioners n = 15, Endocrinologists n = 11, Gynaecologists n = 10) highlighted their concerns about women with PCOS reading poor quality information online, and also how PCOS myths and misinformation are perpetuated on the internet [17].

Women with PCOS brought attention to the negative aspect of reading online comments and how it can lead to distress, feeling isolated, and even anxiety, especially for those trying to get pregnant [44,45,101]. Nevertheless, positive feedback regarding how those with PCOS consider online forums to be a place where they can find support by connecting with others going through relatable struggles, finding anecdotal information was shared in three Australian studies [44,45,102], one in the UK [101], and one in Canada [98].

## 4. Discussion

The results of the present scoping review offer novel insight into the variety, complexity, and connectivity of devices and technology under the umbrella of the IoT, which is currently being used to track, capture, and access PCOS-related information. Interestingly, these are used not just by those with PCOS, but also by clinicians and researchers alike. Of note, the noted surge in publications since 2020, particularly in 2023, underscores a growing relevance and interest in this field. For those with PCOS, the internet and social media emerge as pivotal sources of information through convenience, a sense of privacy, and offering access to peer support networks. However, our findings show that concerns persist about the quality and reliability of online information. Such doubts, which are shared equally by patients and healthcare professionals, are evidenced by the results regarding websites, where the proliferation of PCOS-related content is often tied up to commercial interests and lacks cultural considerations. Similarly, while social media platforms serve as powerful mediums for disseminating PCOS information and fostering peer connections, our results highlight their potential to perpetuate misinformation and overwhelm those with PCOS. Table 2 provides a summary of the pros and cons of using the IoT for PCOS, according to the results of this scoping review.

Mobile apps present a promising avenue to support PCOS management offering features such as menstrual cycle and symptom tracking, education, and support, particularly since women with PCOS voiced a discernible preference for PCOS-specific apps. Wearables and mobile phones can further augment the tracking of health and PCOS-related data. This may further enable personalized monitoring and communication with healthcare providers, thereby contributing towards individualized interventions. The integration of IoT technologies, including AI and machine learning, holds the potential to aid PCOS diagnosis and risk prediction, aimed at reducing delays in diagnosis [103,104]. Our results also illustrate how machine learning algorithms have been leveraged for app development and diagnostic purposes, showcasing their utility as decision-making support tools in clinical practice. The development of an AI-enabled tool capable of integration into electronic health record systems could provide valuable support to healthcare providers with early detection of PCOS [104]. Such advancements can potentially help to address existing gaps in PCOS care and mitigate the dissatisfaction with the diagnosis and management of PCOS reported globally [105,106,107,108].

While our results show that wearables have been used for data collection from participants during trials, there is also the possibility of using wearables for self-management and tracking of symptoms [61,62]. Moreover, it is possible that feedback from wearables could motivate patients with PCOS and keep them on track with their goals. Indeed, a large umbrella review of 39 systematic reviews, including 390 component experimental studies and more than 163,000 participants, found wearable activity-tracker interventions effective in increasing physical activity and supporting modest weight loss [23]. While the authors recommended the use of wearable activity trackers to increase physical activity, they alerted to the fact that most of the included trials were conducted in high-income countries [23]. Therefore, despite promising results, it is crucial to reflect on the socio-economic circumstances of those with PCOS to ensure financial accessibility, cultural sensitivity, and technological literacy while considering the use of wearables or any other technology, such as smartphones and mobile apps.

Mobile apps can support those with PCOS by providing concise information based on scientific evidence, along with a multitude of symptom tracking which can facilitate communication with healthcare providers. However, the multitude of commercially driven apps may be preventing these users from identifying the best available evidence. The AskPCOS app was co-developed by and for those with PCOS and based on the information from the international PCOS guideline [59]. This app and its website include features based on the international PCOS guideline, such as a discussion forum, a PCOS question prompt list for healthcare visits, and advice on lifestyle, menstrual health, weight stigma, fertility, and more [109]. The official website that hosts the international PCOS guideline [110] contains resources for the public, clinicians, and policymakers, including infographics, factsheets, and booklets with PCOS information [111].

The popularity of online PCOS resources has been highlighted by research into the lived experiences of those with PCOS [105]. Turning to internet-based resources and/or mobile apps for information and support has been previously identified in other aspects of women’s health, such as endometriosis [112] and pregnancy [113]. Healthcare professionals have mixed views in relation to patients resorting to the internet [114] and mobile apps [115], with reservations about the sources and accuracy of information, yet highlighting that active patient engagement can support a positive patient-physician interaction [116,117]. Nevertheless, websites and social media constitute the main sources of information for those with PCOS, likely driven by popularity, ease of access, and influencer virality, as shown by the reported 1.8 million average views for PCOS-related content on TikTok [73]. Caution is warranted, since a significant portion of PCOS-related content on social media is created by those who are selling supplements, health coaching sessions, courses, and/or consultations [72,73]. The extent of such conflict of interest was reported to be 45% on TikTok posts and 89% on Instagram posts [73]. However, due to its outreach, social media can be used to combat misinformation and contribute towards ensuring the public can access accurate PCOS information and advice. For example, along with PCOS support groups, some prominent PCOS researchers are also active on social media, particularly during the month of September, i.e., PCOS Awareness Month [70,71]. Their evidence-based posts on social media can contribute towards drowning out the commercial bias and misinformation content. There is, however, a need for culturally appropriate messages [54,92] and to extend the outreach to African, Asian, South American, and non-English speaking European countries [71]. Videos with critically reviewed information curated from national and international guidelines using a combination of illustrations and infographics could be uploaded in a range of languages and promoted through various easily accessible platforms [118,119,120,121].

### 4.1. Gaps in PCOS Care and Education

It is vital to ensure that clinicians are versed in the updated 2023 PCOS guideline [3], have access to any tools that simplify and speed up the diagnostic process, as well as support them to provide patients with the most up-to-date and personalized care needed. The proliferation of AI and Large Language Models, such as ChatGPT [122] and CoPilot [123], opens the possibility of providing a more interactive experience to both patients and healthcare professionals when they search for information about PCOS. Feeding the information from the international PCOS guideline [3] would help to ensure that when interacting with AI tools and chatbots, the answers, advice, and guidance provided would be based on internationally accepted guidelines. Consequently, this may contribute to countering the current trend confirmed in our results of people obtaining information from unreliable sources on websites and social media. The anecdotal use of internet searches for medical information, commonly referred to as “Dr Google” has been reported to facilitate the clinical encounter between the patient and the physician [124,125] without impacting adherence to treatment [126]. Given such a precedent with “Dr Google”, it is likely that similar patient engagement with AI will develop and flourish. While there are initial reports of patient engagement and satisfaction with AI being used in healthcare settings, the human component is still a crucial part of the process [127,128,129]. Studies have shown that chatbots are able to generate quality and empathetic responses to patients [130] and substantially increase referrals [131]. A personalized self-referral chatbot in the UK, geared towards mental health and available to the general population, was able to increase referrals from 6% to 15%, with a particular impact on ethnic minorities (29% increase) [131]. A conversational AI chatbot developed as part of a postpartum support program piloted at a hospital in the USA [128] received high levels of engagement (more than 98% of patients asked at least one question) and accuracy (over 70% of questions answered correctly) [129]. Along with freeing up clinicians’ time to focus on complex cases, preliminary patient feedback also showed a positive impact on health equity, with Black patients statistically more likely to promote the program compared to White patients [129]. Therefore, it is plausible to consider that a chatbot trained on the most up-to-date evidence-based PCOS knowledge from the guidelines could become a valuable tool for patients and clinicians alike.

Developers of a menstrual cycle tracker and PCOS diagnosis app proposed to train a chatbot using neural networks and deep learning to go beyond answering common questions about menstruation by also providing personalized advice and support to its users [65]. An app designed to track menstruation already uses AI algorithms to give the user personalized menstrual and ovulation predictions [56]. This also includes a chatbot functionality capable of engaging in dialogue with the user, based on 18 questions focusing on reproductive and general health [56]. Considering the rapid evolution of AI and machine learning-powered chatbots, there is a potential opportunity to work on the development of chatbots capable of providing evidence-based advice to those with PCOS-related questions.

Finally, our results also highlight how machine learning can play a role as a diagnostic aid [87], and how it can be incorporated into apps capable of further supporting the patient by providing exercise and fitness advice [64] or even a chatbot function [65]. The use of wearables such as activity [78,79] and sleep trackers [77,80], as well as mobile apps for symptom tracking [53,54], can keep clinicians informed about patients’ progress, consequently promoting communication and engagement. People with PCOS have expressed a preference to use an app over a website, particularly if a PCOS-specific app gives them access to evidence-based information, allows them to record symptoms, and offers the opportunity to ask questions to an expert [58]. Such willingness by people with PCOS to ask questions is further supported by their reported engagement with a PCOS chatbot [56]. Along with health and technology literacy, language, socio-economic factors, and cultural background sensitivities, all need to be considered when devising plans to use the IoT for those with PCOS. Therefore, it is crucial to involve patients in the co-creation process [105].

### 4.2. Strengths and Limitations

A robust methodology was followed in the context of this scoping review. The number of results included, comprising a mixture of qualitative and analytical studies, offers a nuanced perspective on the applications of the IoT in PCOS. The contemporary evidence included, coupled with the comprehensive mapping of results across distinct IoT domains, underpins the relevance and currency of our findings. However, we acknowledge the limitations inherent in the selection of databases searched and the effectiveness of our search strategy, which despite being meticulously designed, may have resulted in the inadvertent omission of relevant literature. Cultural aspects that can impact digital literacy, acceptance, access, and use of technology, along with the influence of local factors on various levels of internet availability, should all be noted as potentially impacting on the heterogeneity and generalizability of the findings. Additionally, challenges deriving from the subjectivity of synthesizing qualitative findings and the absence of formal quality assessment, such as the risk of bias and certainly of evidence, underscores the need for a cautious interpretation of our findings.

## 5. Conclusions

The notable increase in publications on PCOS-related IoT underscores a growing potential for machine learning, mobile apps, wearables, and particularly the interconnectivity between IoT technology to facilitate personalized management of PCOS. Overall, our results show that patients, clinicians, and researchers are engaging with the IoT for PCOS-related solutions and further highlight relevant gaps, which should be considered by those using IoT solutions/platforms for PCOS. Given the complexity and ongoing rapid advancements in the relevant technology/IoT, potential recommendations on the use of such IoT solutions require additional high-quality evidence. Work in this field should be rooted in key factors such as accessibility, cultural relevance, and seamless integration into clinical practice. Furthermore, it is paramount to address concerns surrounding the reliability and proliferation of misinformation on social media platforms and online resources. Embracing principles of cultural and socio-economic relevance, technological literacy, and active patient engagement through co-creation will help to harness IoT innovation to bridge gaps in care, foster evidence-based education, and cultivate patient-centered PCOS care that is effective in improving the health of this underserved population.

## Figures and Tables

**Figure 1 healthcare-12-01671-f001:**
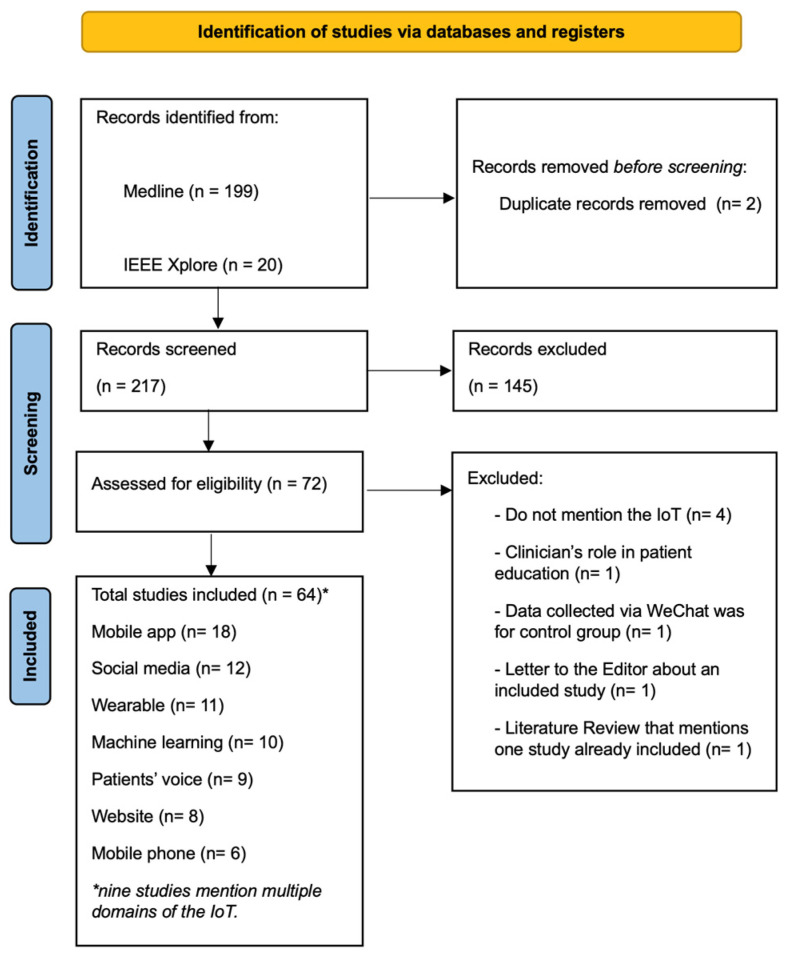
PRISMA Flow diagram of database searches and study screening.

**Figure 2 healthcare-12-01671-f002:**
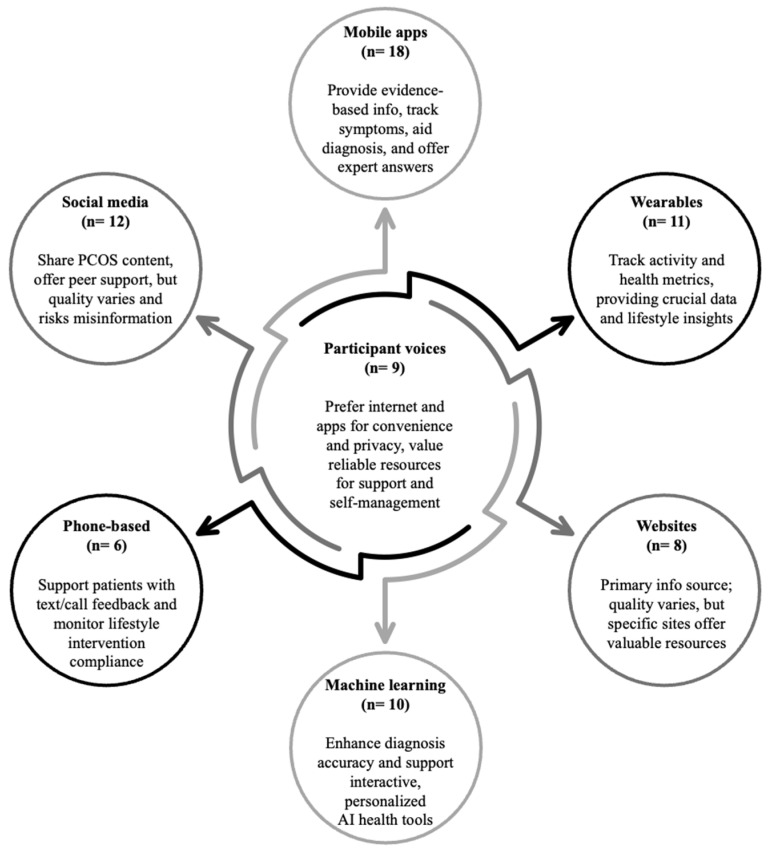
Summary of results: PCOS and the IoT (n = 64, nine studies mentioned multiple domains of the IoT).

**Table 1 healthcare-12-01671-t001:** Themes identified from the included papers reporting on participants’ voices.

Study ID	Source of Information	View on Source of Information	Value	Concern
Cowan et al., 2023 [97]	Internet and social media	Sources often provide inaccurate and ineffective lifestyle advice	Internet and social media are primary sources for diet and activity information	Emphasizes need to increase engagement with qualified health professionals
Ismaylova and Yaya, 2022 [98]	Online support groups	Provides emotional support and a sense of community	Connecting with others facing similar challenges as a source of support	Absence of support before joining online groups led to feelings of isolation and depression
Copp et al., 2021 [17]	PCOS Australia Facebook group (social media)	Peer group may not be helpful as people have unique experiences	Positive experiences were reported from connecting with others	Concerns about potential stigmatization and the anxiety caused by reading others’ negative experiences
Lim et al., 2021 [44]; Ee et al., 2020 [45]	Internet and online support groups	Gap in supportive and specific information for PCOS management	Sense of relatability in online posts	Stories and comments can create anxiety, and it can sometimes be a source of negativity
Holton et al., 2018 [99]	Internet and Facebook (social media)	The internet and online support groups can provide valuable information	Preference for evidence-based information in an accessible format, such as trusted websites and podcasts with health professionals	Government factsheets can lack valuable information and academic sources can be dense
Williams et al., 2016 [100]	PCOS conference and Tumblr (social media)	Useful for healthy meal information and learning about alternative treatment options	Shared experience is valuable with alternative treatment options shared	Fear of side effects from conventional treatments led to the search for alternatives
Holbrey and Coulson, 2013 [101]	Online support group	Helpful to connect with others facing similar challenges	Support group helpful for discussing issues and concerns with people who understand	Reading about others’ severe problems sometimes led to increased anxiety
Avery and Braunack-Mayer, 2007 [102]	Internet	Easily accessible, private, and valuable source of information that allows for multiple information queries	Ability to access a wealth of information at any time, in privacy	Exploring information on the internet might not be suitable for everyone

**Table 2 healthcare-12-01671-t002:** Pros and cons of using the Internet of Things (IoT) for polycystic ovary syndrome (PCOS).

IoT	Pros	Cons
Mobile apps	Facilitates symptom tracking, menstrual cycle monitoring, and education	Concerns about data privacy, security, and the quality and reliability of the information provided
Social media	Useful for disseminating PCOS-related information and increasing awareness	Content quality can be variable and unreliable, with the potential for spreading misinformation
Wearables	Enables detailed symptom monitoring and real-time health data collection	Cost and accessibility issues, along with concerns about data security and patient privacy
Machine learning	Shows promising results in PCOS diagnosis accuracy, risk prediction, and mobile app development	Requires large and diverse datasets, and implementation can be complex and resource-intensive
Websites	Among the abundant PCOS-related content, internet forums provide emotional support and first-hand knowledge sharing between patients	User access limited by digital literacy and accessibility, whilst the content may lack quality and cultural considerations
Phone-based	Provides direct feedback and support, aiding in behavior change and self-management	Limited to user access to technology

## Data Availability

No new data were created for this scoping review. Data sharing is not applicable to this scoping review article.

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
