# Peer review of "Polycystic Ovary Syndrome and the Internet of Things: A Scoping Review"

_healthcare, 2024, doi:10.3390/healthcare12161671_

Round 1
Reviewer 1 Report
Comments and Suggestions for Authors
The authors made a great comprehensive overview of the potential role of the Internet of Things in managing polycystic ovary syndrome.
The concluding statement is interesting, emphasizing the potential of IoT to improve PCOS care.
The manuscript is well-structured, and balanced.
Introduction is clear, even if there were a lot of references in it.
Methodology is well explained, however : mention of wearables and machine learning is intriguing, I would suggest authors to add more details on subtypes of Applications and IA resources or wearables.
Add these subtypes and define them according to the studies that used them.
The inclusion of participants' perspectives is valuable. However, more specific feedback or common themes from participants' experiences with IoT tools could be highlighted to give deeper insights into user acceptance and challenges.
The authors noted concerns about the quality and reliability of social media content and websites. It would be beneficial to briefly mention any criteria for evaluating trustworthy sources to guide clinicians or patients.
I noticed that the majority of these studies are from india. Authors ahve to address the heterogeneity of the studies based on access to the internet, it's important to recognize the varying levels of internet availability and digital literacy among different populations (urban / rural )
Reflect on cultural differences in the acceptance and use of technology. Some cultures may have reservations about using certain types of technology for health management, impacting the generalizability of study findings.
Comments on the Quality of English Language
minor editing
Author Response
Reviewer-1
Comments: The authors made a great comprehensive overview of the potential role of the Internet of Things in managing polycystic ovary syndrome.
The concluding statement is interesting, emphasizing the potential of IoT to improve PCOS care.
The manuscript is well-structured, and balanced. Introduction is clear, even if there were a lot of references in it.
1) Methodology is well explained, however : mention of wearables and machine learning is intriguing, I would suggest authors to add more details on subtypes of Applications and IA resources or wearables. Add these subtypes and define them according to the studies that used them.
2) The inclusion of participants' perspectives is valuable. However, more specific feedback or common themes from participants' experiences with IoT tools could be highlighted to give deeper insights into user acceptance and challenges.
3) The authors noted concerns about the quality and reliability of social media content and websites. It would be beneficial to briefly mention any criteria for evaluating trustworthy sources to guide clinicians or patients.
4) I noticed that the majority of these studies are from india. Authors ahve to address the heterogeneity of the studies based on access to the internet, it's important to recognize the varying levels of internet availability and digital literacy among different populations (urban / rural). Reflect on cultural differences in the acceptance and use of technology. Some cultures may have reservations about using certain types of technology for health management, impacting the generalizability of study findings.
Reply: We thank the reviewer for the time dedicated to reviewing our paper and for the positive feedback and helpful comments/suggestions. Accordingly, we have made the following revisions based on the provided comments:
1) We added more details in the methods section to address the comment about the IoT subcategories of mobile apps, artificial intelligence, and wearables (please see lines 136-143). Further, in the results section (please see lines 178-182), we reference the seven individual domains of IoT, their related included studies, and how some studies mention multiple domains of the IoT (for example, one study uses mobile apps, wearables, and machine learning). Figure 2 provides a visual representation of the different IoT domains in the context of PCOS, and the table of results (supplementary file S3) shows all the data extracted from the studies included.
2) We added a new table with themes identified from the included papers reporting on participants’ voices (please see Table 1 - lines 407-410). We also provide a dedicated table of results highlighting the participants’ voice from their experiences of using IoT for PCOS (please see supplementary file S4).
3) The criteria used in the included studies to assess the quality and reliability of social media content and websites have been reported in the Table of Results (supplementary file S3), and includes tools such as DISCERN: 16-item DISCERN questionnaire, an instrument for judging the quality of written consumer health information on treatment choices, and EQUIP: 36-item EQIP (ensuring quality information for patients) tool.
4) We have added a relevant point to the limitations (please see lines 591-594) to address the useful reminder about heterogeneity and generalizability of the findings.
In addition, across the results we include examples of culturally sensitive studies, e.g. from Saudi Arabia: Alotaibi & Shaman, 2020 [ref 40] and Alotaibi & Alsinan, 2016 [ref 41] - “Mobile PCOS Management and Awareness System for Gulf Countries”, and how despite restricted internet and social media access in China, the platform WeChat is being used to emulate solutions seen in studies originating from other locations.
Overall, the full results were fairly distributed between USA (n= 11, 17.2%), Australia (n= 10, 15.6%), UK (n= 8, 12.5%), seven (10.9%) each from China and India, and four (6.3%) from The Netherlands. There were also two studies (3.1%) each from Brazil, Canada, Croatia, New Zealand, Korea, and Saudi Arabia, and one study (1.6%) each from France, Greece, Jamaica, Poland, and Turkey (lines 189-194).
In line with the provided suggestion, we also emphasize in our discussion that “despite promising results, it is crucial to reflect on the socio-economic circumstances of those with PCOS to ensure financial accessibility, cultural sensitivity, and technological literacy while considering the use of wearables or any other technology, such as smartphones and mobile apps” (lines 488-491). Furthermore, in our conclusion, we emphasize the need for “Embracing principles of cultural and socio-economic relevance, technological literacy, and active patient engagement…” (lines 612-613).
Reviewer 2 Report
Comments and Suggestions for Authors
Dear Editor
Your review is practical for management of this common condition.
1-what is the final conclusion of using IoT in PCO patients based on your study
2-In order to sumerize the result it is better to show the summary of included studies in table.
3-what is the Con's and pros of using IoT for managing pco?
4_what platform do you recommend for patients according to six domain of IoT in your study.
Author Response
Reviewer-2
Comments: Your review is practical for management of this common condition.
1-What is the final conclusion of using IoT in PCOS patients based on your study?
2-In order to summarize the result it is better to show the summary of included studies in table.
3-What is the cons and pros of using IoT for managing PCOS?
4-What platform do you recommend for patients according to six domain of IoT in your study.
Reply: We thank the reviewer for the time dedicated to reviewing our manuscript and the helpful feedback/comments. Please find below the responses addressing the provided questions, comments, and suggestions:
1) A plausible final conclusion that can be drawn from the results of our study is that the notable increase in publications on PCOS-related IoT underscores its growing potential (lines 602-604) and therefore “advancing evidence-based, culturally sensitive, and accessible IoT solutions can enhance their potential to transform PCOS care, address misinformation, and empower women to better manage their symptoms” (Abstract, lines 40-42).
2) As suggested, a full table of results for the six domains of IoT is provided in supplementary file S3 (please see supplementary file S3), as well as a separate table for the participants’ voice (please see supplementary file S4), whilst Figure 2 (line 180) provides a visual representation and summary of results. We also added a new table (please see Table 1, lines 407-410) which summarizes the themes from the participants’ voices.
3) We added a table with pros and cons of using the IoT for PCOS (please see Table 2, lines 463-466).
4) We added more details to the conclusion in order to address the question about recommendation of a specific IoT platform (please see lines 604-609).
Reviewer 3 Report
Comments and Suggestions for Authors
The authors have raised an interesting and relevant topic in the context of PCOS syndrome. It is well known that control and access to information, as well as a multidisciplinary approach, is crucial for patients with such a syndrome. Increasing patient awareness plays a very important role, and in today's world of increasing digitization, access to new possibilities as for example applications can be quite an advantage, although it may involve some negative consequences in the case of unmeritorious content. Hence, the topic presented is very timely and significant for the access to information of patients with PCOS, but also for medical personnel, who should create awareness and educate especially young patients on the use of available content on this condition.
Comments on the Quality of English LanguageNo comments
Author Response
Reviewer-3
Comments: The authors have raised an interesting and relevant topic in the context of PCOS syndrome. It is well known that control and access to information, as well as a multidisciplinary approach, is crucial for patients with such a syndrome. Increasing patient awareness plays a very important role, and in today's world of increasing digitization, access to new possibilities as for example applications can be quite an advantage, although it may involve some negative consequences in the case of unmeritorious content. Hence, the topic presented is very timely and significant for the access to information of patients with PCOS, but also for medical personnel, who should create awareness and educate especially young patients on the use of available content on this condition.
Reply: We thank the reviewer for the time dedicated to reviewing our paper and for the positive feedback.